# The Relationship between Corporate Social Responsibility on Social Media and Brand Advocacy Behavior of Customers in the Banking Context

**DOI:** 10.3390/bs13010032

**Published:** 2022-12-29

**Authors:** Shijiao Huang, Xu Hai, Nawal Abdalla Adam, Qinghua Fu, Aqeel Ahmad, Daniela Zapodeanu, Daniel Badulescu

**Affiliations:** 1School of Intercultural Studies, Jiangxi Normal University, Nanchang 330022, China; 2Graduate School of Business, Universitiy Tun Abdul Razak, Kuala Lumpur 50400, Malaysia; 3Department of Business Administration, College of Business & Administration, Princess Nourah Bint Abdulrahman University, Riyadh 11671, Saudi Arabia; 4Department of Business Administration, Moutai Institute, Zunyi 563000, China; 5Faculty of Management Sciences, University of Central Punjab, Lahore 54000, Pakistan; 6Department of Finance and Accountancy, Faculty of Economic Sciences, University of Oradea, 410087 Oradea, Romania; 7Department of Economics and Business, Faculty of Economic Sciences, University of Oradea, 410087 Oradea, Romania

**Keywords:** CSR, emotions, customer behavior, social media, brand advocacy

## Abstract

This research study aims to investigate the relationship between corporate social responsibility (CSR)-related communication on social media and brand advocacy behavior of retail banking customers in a developing country. This study also proposes a dual mediating mechanism of customer engagement and customer-company identification in the above-proposed relationship. The data were collected from retail banking customers with the help of a self-administered questionnaire (*n* = 356). To test the hypothesized relationships, a theoretical model was developed in this study. For hypothesis testing, we used the structural equation modeling (SEM) technique in AMOS software. The empirical analysis results confirmed our theoretical assumption that the manifestation of CSR-related communication on social media by a bank significantly influenced the advocacy behavior of retail banking customers. Our study also confirmed the mediating function of customer engagement and customer-company identification. The findings of this study offer different implications for the banking sector. For example, our study highlights the critical role of CSR-related communication on social media for meaningful customer–brand relationships by promoting the advocacy behavior of customers.

## 1. Introduction

Social media, since its emergence, has been recognized as a powerful communication medium for contemporary businesses in all areas and industries. According to a recent estimate, more than half of the world’s population (56%) uses social media actively. At the same time, during the last decade, social media consumer traffic has increased from 970 million to almost 4.5 billion [1]. Moreover, since 2015, social media users have been growing worldwide with an annual rate of more than 12%, which is enormous. Considering the rising importance of social media, businesses worldwide have been actively using it for different marketing, communication, and customer relationship purposes [2,3]. The interactive, participative, and transparent atmosphere of social media makes it an attractive communication forum for organizations to develop a meaningful customer-company relationship [4]. In the same vein, organizations in the current age have been using social media to communicate with customers about corporate social responsibility (CSR) activities. Global brands such as Starbucks and TOMS not only use social media for their product/service information, but they also use social media as a forum to let the customer know about their sustainability engagement to benefit all stakeholders, including the ecosystem and biosphere.

Another emerging trend for contemporary businesses in the recent era is escalating importance of customers to the success of an organization. Dynamic business environment, digital technologies, cut-throat rivalry, and many other factors force contemporary organizations to build meaningful customer relationships [5]. Undoubtedly, without the support of customers, it is almost impossible for a business to excel. Perhaps, this is why contemporary organizations are taking different initiatives to influence customer behavior positively. For example, different models have been proposed to influence customer loyalty and purchase intentions [6,7]. Even in a CSR framework, discussion exists on how an organization’s different CSR-related initiatives can positively determine customer behavior [8,9].

Although the role of CSR from a customer perspective has been highlighted recently at different levels, engaging customers in CSR-related communication on social media has remained an under-researched area with some exceptions [10,11]. Considering the rising importance of social media and the potential role of CSR in influencing customer behavior, it is very important to explore how CSR-related communications of an organization can engage customers meaningfully with a brand.

Another critical issue in most of the previous CSR-customer behavior studies lies with the investigation approach. To date, most of the CSR investigations have been carried out to see the influence of CSR activities of an organization on either customer loyalty [12,13] or purchase intentions [14,15]. We do not tend to underestimate such investigations because, after all, having loyal customers with positive purchase preferences attracts every organization. Nonetheless, we want to extend the discussion on the CSR–customer behavior relationship by highlighting an important customer behavior regarded as customer advocacy behavior (ADOC). By definition, ADOC is a behavior of customers in which they not only promote a brand, but also defend it against detractors [16]. Indeed, the literature recognizes ADOC as a kind of behavior that is more important than loyalty and purchase preferences [17,18] because loyal customers continue to purchase from their favorite brand; however, unlike advocates, they are less likely to promote a brand among their social circles. Considering the seminal importance of customers as brand advocates, contemporary organizations are taking different initiatives to convert their customers into brand advocates. In this respect, some recent scholars have argued in favor of CSR to influence ADOC [19,20]. However, such sparse studies indicate a dire need to conduct more research in this area.

Conceivably Glavas [21] and Aguinis and Glavas [22] were among the first to highlight the role of different psychological factors as mediators to understand the underlying mechanism of how and why CSR influences individual behavior positively. These authors believed that the manifestation of different psychological factors as mediators justifies why CSR can significantly determine/influence/shape a certain behavior of individuals. This is why various scholars have proposed different psychological factors as mediators or moderators to understanding CSR–individual behavior relationships. In this regard, an increasing body of knowledge has realized the role of human emotions in influencing certain behavior. Indeed, the existing body of knowledge indicates that human emotions drive customers’ loyalty and advocacy behavior [23]. Knowing the seminal role of human emotions in influencing a certain behavior, companies in the current age desire to build an emotional connection with customers. Organizations realize that shared values (CSR, for example) may be critical for generating emotional connections that drive customers’ trust in an organization. In this respect, past research indicates that CSR positively influences customer emotions, such as engagement (CE), which then influences customer outcomes. Indeed, the mediating role of CE, as an outcome of CSR, was highlighted previously [13,24]. However, the mediating role of CE in a CSR framework to influence ADOC was not highlighted previously. Because there is a role of emotions in forming a certain behavior and because CSR positively influences human emotions, it is worthwhile to investigate the mediating role of CE between CSR and ADOC.

Similarly, another important psychological factor related to customer emotions is customer-company identification (CCI), which is defined as a customer’s psychological feeling of belongingness with an organization [25]. Recently, in the relationship marketing context, fostering CCI has become an important business imperative because research indicates that customers with an improved level of identification with an organization not only show a higher level of satisfaction but such customers also become loyal to a particular brand [26,27]. At the same time, the existing body of research indicates that customers with a higher level of CCI are expected to promote a company toward excellence [28]. Although the mediating role of CCI to influence customers’ outcomes has remained in academic discussion for a long time, the mediating role of CCI to influence ADOC in a CSR framework was not highlighted previously. Therefore, this research tends to bridge this knowledge gap by exploring the mediating effect of CCI between CSR and ADOC.

In precise, our research study has three specific objectives. Firstly, our study aims to explore how CSR-related communication of an organization on social media engages customers meaningfully. As identified earlier, although the role of CSR from a customer perspective has been highlighted recently, engaging customers in CSR-related communication on social media has remained under-researched. Considering the rising importance of social media and the potential role of CSR in influencing customer behavior, it will be interesting to investigate how CSR-related communications of an organization can engage customers meaningfully with a brand.

Secondly, our study aims to close a critical knowledge gap in previous CSR–customer behavior literature. To this end, most of the CSR investigations have been carried out to see the influence of CSR activities of an organization on either customer loyalty or purchase intentions, and the relationship between CSR and ADOC has remained an understudied area. Considering the seminal importance of customers as brand advocates to the success of an organization, it will be worthwhile to investigate how CSR-related communication of an organization on social media can influence ADOC.

Lastly, our research aims to enrich the existing literature on CSR by highlighting the role of human emotions in influencing customer behavior. Specifically, we introduce CE and CCI as mediating variables to explain the underlying mechanism of why CSR-related communication of an organization on social media influences ADOC. Because an increasing body of knowledge has already realized the role of human emotions in influencing certain behavior, it will be worth investigating the CSR–ADOC relationship by understanding the mediating mechanism of CE and CCI.

This research chooses the banking sector of Pakistan to test the hypothesized relationships. This sector was purposefully selected for three specific reasons. First, the banking sector is one of the service segments which operates under a standard operating procedure (SOP) imposed by the regulator (the state bank of Pakistan in the current case). The SOPs in this segment are the same for every banking firm, which means that this industry faces the issue of competitive convergence, implying that very limited room is left for a certain bank to search for a stable base of competitive advantage. Previous researchers have also indicated this limitation of the banking sector [29,30,31]. In this respect, a well-planned CSR strategy to engage customers on social media may be helpful for a particular bank from the perspective of competitiveness because a bank, in response to CSR, can convert customers as brand advocates who play a significant role in the success of a bank. A second reason for choosing the banking sector for this study lies in the human nature of this industry. Unlike the manufacturing segment, prior testing and experience are not possible in many service segments dependent on humans, making the role of customers as advocates critical. Past research has indicated that, in the service segment, especially in the case of human-dependent segments, customer endorsement or recommendations play a crucial role in the success of a brand [32,33,34]. Hence, it is relevant to influence the debate on ADOC in a CSR framework from the banking industry’s perspective. Another reason for considering the banking sector is that the banking sector in Pakistan is one of the pioneering segments with various CSR-related programs and uses different social media forums to communicate its CSR activities with the external community, including customers.

The theoretical roots of this study are grounded in social identity theory (ST). Proposed by Polish social psychologist Henri Tajfel, the major tenet of this theory is that a particular person tends to identify themself with a social group that enhances their self-esteem [35]. Tajfel believed that individuals consider a social group superior to whom they belong; thus, they attempt to enhance the image and status of their social group. From a marketing communication perspective, when the self-concept of customers is aligned with the image of a particular brand, it generates positive emotions among customers to strongly identify themselves with a brand they belong to [36]. Reflecting on the process of social identity in this study, we argue that, when customers see the CSR-related communication of a particular bank on different social media forums, they appreciate such social engagement of an ethical banking organization in the larger interest of all stakeholders. Consequently, customers are expected to identify themselves with socially responsible brands strongly. Thus, ST can explain the logic of why customers strongly identify themselves with a socially responsible brand.

All in all, this research study tends to enrich the existing body of knowledge in different ways. First, this is one of the limited CSR studies from the perspective of ADOC. In this respect, as specified earlier, most of the CSR–customer behavior studies were conducted either from a loyalty [37,38] and satisfaction [39] perspective or from a perspective of purchase preference [14]. Because the literature rates ADOC beyond loyalty and purchase preferences, it is worthwhile to see how customers’ CSR engagement on social media with a specific brand can promote their ADOC. Second, this study is one of the pioneering studies in the domain of CSR which attempts to highlight the seminal role of human emotions from a behavioral perspective by proposing the dual mediating roles of CE and CCI to understand the underlying mechanism of why engaging customers in CSR-related communication on social media can foster their advocacy behavior. This perspective, according to our best knowledge, has not previously been highlighted. Third, most of the previous studies on CSR–customer behavior relationships were conducted in the context of developed countries [40,41]. Considering the cultural and context-specific nature of CSR, such studies from the perspective of developed countries may not reflect the case of developing countries because developed and developing countries are dissimilar in many ways with respect to infrastructure, market complexity, customer awareness, etc. Therefore, more studies are required from the standpoint of developing countries.

## 2. Literature Review

Past research on CSR–customer behavior has revealed that CSR activities of an ethical organization can influence different outcomes on the part of individuals. Specifically, it was mentioned in the prior body of knowledge that CSR can influence different extra role behaviors of individuals, including customers’ citizenship behavior [42,43] and pro-environmental behavior [44,45]. From the perspective of marketing communication, past research highlights the role of CSR-related communication of an organization on social media to influence different communicative behaviors of customers. For example, Bialkova and Te Paske [46] indicated that communicating with customers about the CSR activities of an organization on social media could significantly influence the positive communicative behavior of customers, including electronic word of mouth. The same kind of observation was presented by Chu and Chen [10], who argued that social media is an effective forum to engage customers in CSR communication for a brand meaningfully. Recent research by Castro-González et al. [47] mentioned that the CSR activities of an organization can derive ADOC for a particular brand. Indeed, scholars have argued that CSR has a seminal role in shaping/influencing and modifying different behavioral intentions of customers [48,49]. From the perspective of social media, the existing discussion on CSR–customer behavior identifies social media as an effective forum to inform customers about the CSR-based actions of an organization. For example, Gupta et al. [50] posited that a particular banking institution’s CSR-related information significantly enhances customers’ purchase intentions. In a similar fashion, Chu and Chen [10] mentioned that CSR-related communication of an organization positively influences positive word of mouth and purchase intentions of Chinese customers. CSR scholars such as Ahmad et al. [4] believe that, when an organization uses different social media forums to communicate its different CSR-based actions with the customers, such communication significantly predicts the loyalty intentions of customers. Indeed, the interactive interface of different social media forums enables customers to be engaged with a brand [10]. Companies share their CSR activities on social media and initiate a dialogical conversation with customers who share their views not only with the company but also within their social circles. All this process effectively engages customers with a brand, as indicated by previous CSR scholars [51,52]. In this respect, different banking institutions in Pakistan share their CSR activities on social media. Specifically, banks are involved in different CSR activities, including community education, health, environment-related efforts, donations for the deprived segment of society, economic development, and best practices [53,54]. Moreover, concerning ST, we argue that there is a role of CSR activities in an organization to influence ADOC. Specifically, the socially responsible image of an organization generates positive emotions among customers, and they are expected to show a higher level of motivation to foster the social image (CSR-related in the current scenario) of the group they belong. More specifically, when customers observe the CSR engagement of a socially responsible organization on social media, they not only show respect for such organizations but also communicate such noble engagement of an ethical organization to others, including their friends, peers, and family members. This leads to the following hypothesis:

**Hypothesis** **1** **(H1).***Engaging customers in CSR-related communication on social media can influence their advocacy behavior for a particular brand*.

The term CE has received increasing attention from scholars in recent years. In particular, the literature on marketing communication has largely discussed different benefits that an organization can reap from a higher level of CE [55,56]. Specifically, the available body of knowledge recognizes the potential benefits of CE to establish and sustain a meaningful competitive advantage [57,58]. Additionally, the literature also rates CE as an important predictor to influence an organization’s overall performance and business excellence [59]. At the same time, past researchers have mentioned that customers with an improved level of engagement exhibit better loyalty [60], emotional bonding [61], satisfaction [62], trust [63], and commitment [64,65,66]. Moreover, CE has been related to influence different communicative behaviors of customers, e.g., brand referrals, customer endorsement, and positive word-of-mouth enhancement [67]. Although there exist various definitions of CE, we tend to refer to the definition by Hollebeek et al. [68], who contended that CE refers to a specific psychological state of a customer with a focal agent (a specific organization or brand) in response to interactive, co-creative customer experience with the focal agent. Hollebeek and colleagues further stated that CE includes emotional, rational and behavioral engagement of customers with a particular brand/product/organization. From the perspective of CSR, the existing literature shows that contemporary customers expect modern businesses to be engaged in different CSR-related activities [69]. Indeed, CSR has emerged as an important business imperative over the recent past, and organizations have started to give it a special place in their communication strategies with the external community, e.g., communicating different CSR activities with different stakeholders on social media (Facebook, Twitter, etc.) and showing CSR spending in annual reports.

Especially with the rise of social media, contemporary customers tend to develop a social consciousness with businesses because they are interested to see how they are helping others by being the purchasers of a socially responsible organization [70]. Mishra and Modi [69] believed that modern customers are passionate about seeing how their relationship with a brand will make this planet a better and sustainable place for future generations. Specifically, Euromonitor’s Top 10 Global Consumer Trends report for 2015 showed that consumption by contemporary customers is significantly driven by heart, implying that customers are making consumption choices derived from their positive impact on the ecosystem and biosphere [67]. In this respect, when customers see that the brand they belong to shares different CSR-related communications on different social media forums, they are self-motivated to support such socially responsible brands by promoting it in their social circles. From the perspective of ADOC, a recent study by Bilro et al. [57], CSR-related communication on social media influences CE positively, which then induces ADOC. These authors further stated that CE, as an outcome of CSR, is an important predictor of converting customers as brand advocates. In a nutshell, the meditating role of CE to influence different customers’ outcomes has been mentioned in the prior literature at many levels [24,71]; because the existing body of knowledge mentions that CSR relates to an improved level of CE [72], we propose the following hypotheses:

**Hypothesis** **2** **(H2).***CSR-related communication on social media can influence the engagement level of customers*.

**Hypothesis** **3** **(H3).***Customer engagement significantly mediates between CSR and advocacy behavior of customers*.

Customers are identified as important stakeholders for an organization that, in recent times, have very high expectations toward contemporary businesses to be ethical by performing different CSR-related responsibilities [73]. With respect to ST, customers are expected to identify themselves with a brand based on some specific characteristics, e.g., the socially responsible behavior of an organization is something that inculcates positive feelings among customers to strongly identify themselves with a brand [74]. Moreover, individuals are expected to categorize themselves into different social categories. According to Fatma et al. [73], this social categorization process is important for two reasons. First, it indicates the importance of the social environment for individuals to define themselves with respect to others. Second, most importantly, social categorization is important for individuals for personal identification with a focal agent to justify the reason for this social identification. In this respect, the literature argues that an ethical organization’s CSR engagement justifies customers why they should develop a strong identification with a socially responsible brand [75].

Moreover, the social commitment of an ethical brand urges customers to develop strong CCI. From an emotional perspective, the socially responsible behavior of a brand motivates customers to relate to such a brand. Explaining this phenomenon further, Scott and Lane [76] indicated that the emotional bonding between customer and company creates an emotional pull among the customers, ultimately promoting them to a higher level of CCI.

In this vein, the increasing importance of social media has further facilitated the available literature on marketing communication to explain how customer engagement on social media with a brand influences their CCI [77,78]. Indeed, the emergence of social media enables customers to make interactive and effective communication with a brand, which was impossible in the conventional communication medium. Fernández et al. [79] mentioned that the manifestation of social media had increased the effectiveness of CSR for an organization because, by using social media as a communication forum, an organization can effectively engage customers in CSR-related communication, which ultimately improves their CCI. Additionally, communicating CSR activities with customers on social media platforms may significantly engage customers emotionally with a brand, which then influences their CCI. In this regard, the past literature has strongly endorsed the mediating effect of CCI to shape/influence different customers’ outcomes, especially extra role outcomes. Even from the standpoint of CSR, there are several studies exhibiting that there is mediating role of CCI to influence different behavioral intentions of customers [80,81]. In line with this literature stream, we argue that CSR communication on social media enhances CCI, creating a mediating effect to influence ADOC. The conceptual model is presented in Figure 1. Therefore, we propose the following hypotheses:

**Hypothesis** **4** **(H4).***CSR-related communication on social media can influence customer company identification*.

**Hypothesis** **5** **(H5).***Customer company identification mediates between CSR and advocacy behavior of customers*.

## 3. Methodology

### 3.1. Study Sector and Data Collection

The banking system in the country is managed by a regulator, which is the State Bank of Pakistan. There are two major banking streams in this South Asian nation, including conventional and Islamic banking. Conventional banking follows regular interest-based banking, whereas the Islamic banking stream operates under an interest-free banking context. The conventional banking system is the dominating banking system in Pakistan which has been operating since the existence of this country on the world’s map (1947). According to a recent estimate, although Islamic banking has been growing in the country, it still only captures around 20% share of the total banking industry [82]. Habib bank limited (HBL) is the largest banking institution representing conventional banking, and Meezan bank (MB) is the pioneer Islamic bank in the country. Because this study intends to investigate the relationship between CSR and ADOC, we selected the five largest banks operating in Karachi and Lahore. The former is the capital of Sindh province, whereas the latter is the capital of Punjab province. The reason for selecting these two cities for this study lies in the fact that these cities are the metropolitan cities of Pakistan where a multimillion population resides. Moreover, almost all banking institutions have multiple branches in these two cities, and, in most cases, even head offices and zonal offices also operate in these two cities. The five sampled banks in this survey were HBL, MB, United bank limited (UBL), Allied bank limited (ABL), and National bank of Pakistan (NBP). The reason to consider these banks is that all these banks have different designated CSR programs, and they communicate CSR-related information with the external community on different social media forums regularly (for example, these banks have established separate pages on Facebook, Twitter, etc. to communicate CSR activities with different stakeholders).

To collect the data from banking customers, we devised a purpose-built strategy in which the customers were requested to fill out this survey questionnaire (printed version following a paper-pencil method) when they were leaving a specific branch of a bank or they were leaving an ATM facility. This strategy was designed due to two specific reasons. First, this strategy enabled us to maintain direct contact with real banking customers. Second, this strategy is also useful for collecting the data because it does not disturb any banking operations of a bank because customers are approached outside the premises of a particular bank. Earlier researchers such as Raza et al. [31] and Sun et al. [30] also argued in favor of this data collection strategy from banking customers. Screening questions were asked from every respondent to ensure whether he or she has the basic knowledge of CSR and actively uses different social media networking sites.

The questionnaire statements were adapted from multiple published and reliable sources. The experts also assessed these statements to ensure that these statements were appropriate to conceptualize a particular variable in light of the current study’s context [83,84]. Additionally, we followed the ethical protocols given in the Helsinki Declaration during the data collection process [85,86,87,88]. Specifically, the data were collected during April and May 2022. The sample size in this study was estimated using an a priori sampling calculator specially designed to estimate a possible recommended sample size for structural equation modeling [89]. The study-specific estimation of this application, accompanied by different other benefits, is among the important reasons making this calculator a contemporary tool for sample size estimation [90,91]. On the basis of some specific inputs, this calculator indicated that the possible sample size recommended for this study should be around 342. Considering that survey studies do not produce an ideal response rate, we intentionally distributed 500 questionnaires among different banking customers initially to get as close as possible to the recommended sample size. As expected, we did not receive all questionnaires from the customers, as they returned 388 fully filled questionnaires, making the initial response rate close to 78%. These filled questionnaires were scrutinized before making them a part of the finalized dataset. Specifically, the questionnaires with missing information and cases identified as outliers were removed (32 cases). Table 1 and Table 2 include detailed information on data cleaning statistics. Please refer to Appendix A for variable items used in this survey

### 3.2. Measures

We used a five-point Likert scale in this study. Specifically, there were four variables: CSR, CE, CCI and ADOC. We adapted the items to measure CSR-related communication on social media from Chu and Chen [10], who modified the original scale of Gil de Zúñiga et al. [92] in the context of social media. There were five items on this particular scale. Example items include “I post personal experiences related to my bank’s CSR activities on social media” and “I post or share thoughts about my bank’s CSR activities on social media”. The inter-item consistency for this scale was 0.84, which was significant and positive. To measure CE, we used a four item scale developed by Hollebeek et al. [68] to measure the emotional aspect of CE. One sample item from this scale was “I feel very positive when I use the products/services of my bank”. The reliability analysis showed a significant value of 0.78 for this variable.

To conceptualize CCI, we used five items given in the study of Eberle et al. [93], which included a sample item, “I have a sense of connection with my bank”. The reliability stats showed a significant value of 0.87 for this variable. Lastly, the variable of ADOC was conceptualized using the scale of Melancon et al. [94], who introduced a four-item scale to measure customers’ perception regarding their advocacy behavior for a particular brand. A sample item from this scale was “I would defend my bank on social media to others if I heard someone speaking poorly about it”. A significant reliability value of 0.74 was observed for this variable.

### 3.3. Social Desirability and Common Method Bias

In developing different methodological-related protocols, we considered how to minimize social desirability and common method bias (CMB) issues. We took different theoretical measures during data collection regarding the social desirability issue. For example, we ensured that the question statements were simple and easy to understand for the respondents. Similarly, we also informed the respondents that their true responses were very important in this survey; hence, they should carefully record a specific response without thinking that it is bad or undesirable. In the same vein, we randomly presented the statements of a variable in the questionnaire so that the respondents could avoid any intended sequence in answering the questions [95,96].

To deal with the issue of CMB in this study, we performed the famous common latent factor test (CLFT). During this process, we developed two measurement models. One was the originally proposed four-factor measurement model, and the other was produced by introducing a common latent factor (CLF), which was allowed to influence every observed variable in this study (a total of 18). To detect any manifestation of CMB, we compared the standardized regression weights of both models. We revealed that the regression weights of both models slightly differed, but these differences were not significant (all <0.2) [97,98,99]. This implies that including a CLF in the measurement model did not produce any significant variance, thus confirming the non-criticality of any CMB issue. The sociodemographic information of the respondents revealed that mostly male respondents contributed to this survey (68%). The ages of most respondents were between 18 and 45 years (almost 89%). The education level of informants varied between intermediate (12%) and master (43%).

## 4. Results

### 4.1. Preliminary Data Analysis

In the initial data analysis phase, we conducted different statistical tests to verify the validity, reliability, goodness of model fit, and correlation analysis. The validity of all variables was verified by calculating the value of average variance extracted (AVE) for each variable separately. The following mathematical expression (Equation (1)) helped us calculate AVEs for this study’s four variables:(1)$$\mathrm{AVE}=\frac{\sum_{\dot{i}=1}^{k}\lambda_{i}^{2}}{\sum_{\dot{i}=1}^{k}\lambda_{i}^{2}+\sum_{i=1}^{k}.var\left(\varepsilon i\right)}.$$

We revealed that AVEs varied from 0.60 (ADOC) to 0.65 (CSR). Specifically, all values exceeded the minimum threshold level (0.5) [100,101,102]. This confirmed the statistical significance of AVEs in all cases. This further confirmed that the convergent validity for CSR, CE, CCI, and ADOC was significant. Hence, the items of a particular variable converged to their respective factor compared to other factors.

In a similar fashion, we used the following mathematical expression to evaluate the composite reliability of a variable:(2)Composite reliability=((∑λi)2)/(∑λi)2+∑var(εi)).

We observed that the composite reliability values for CSR, CE, CCI, and ADOC were greater than 0.7 [103,104,105], statistically proving that the reliability values were significant. Indeed, these values varied from 0.85 (ADOC) to 0.90 (CSR). Hence, the validity and reliability analysis showed a statistical significance in all cases. Similarly, the factor loadings of all variables were also significant (>0.5 and ideally >0.7) [106,107]. Specifically, the factor loadings varied from 0.69 to 0.93. Table 3 can be observed for further detail.

The goodness of model fitness was the next step in preliminary data analysis. Indeed, this step was important to statistically establish that the originally hypothesized model in this study (model 1) superiorly fits the statistical dataset. For this purpose, we prepared four measurement models (1, 2, 3, and 4). Among these models, we observed that a one-factor model showed poor model fit statistics (RMSEA = 0.119, *χ*^2^/*df* = 6.84, GFI = 0.64, TLI = 0.60, IFI = 0.62, CFI = 0.67), implying that there was a poor fit between theory and statistical data. Models 2 and 3 produced mixed results (see Table 4); however, when compared to model 1, we observed that this model was superior in all respects (RMSEA = 0.066, *χ*^2^/*df* = 2.92, GFI = 0.94, TLI = 0.92, IFI = 0.94, CFI = 0.95). This confirmed that there was an exemplary fit between theory and data for model 1.

Lastly, we conducted a correlation test in preliminary data analysis. This correlational analysis was important to see the nature of associations between different variable pairs. Similarly, the correlational values may provide important initial insights toward the significance of hypotheses statements. In this regard, we found that the values of correlations (*r*) were positive and significant in all cases. As an exemplary case, we found that *r =* 0.56 between CSR and CE, which was significant (*p* < 0.05). Similarly, all other cases were significant, as can be seen in Table 5. We also reported divergent validity statistics in the same table (bold values). The divergent validity test showed that all values were significant, indicating that the items of one variable were not similar to the statements of another variable.

### 4.2. Main Data Analysis

After performing different statistical tests and confirming the validity and reliability of all variables, we tested the hypothesized relationships by performing structural equation modeling (SEM), a second-generation advanced-level data analysis technique especially important to analyze complex research models [70,108,109]. In this regard, we considered AMOS software to draw a structural model to validate our hypotheses. Before running the main analysis in AMOS through developing a structural model, we examined our dataset to fulfil initial data analysis criteria. For example, we checked the normality of our data. For this purpose, we checked the skewness and kurtosis values which were within the significant ranges.

The structural model was evaluated in two phases to analyze the hypotheses in this study. In the first phase, we tested the direct effects of different hypothesized relations. This direct effect analysis was performed to establish or reject the statistical significance of H1, H2, and H4. We summarize the results of this analysis in Table 6. As per the results, it was revealed that H1, H2, and H4 were all statistically significant. To arrive at such a conclusion, we observed beta values, T-statistics, confidence intervals (CI), and *p*-statistics values [110,111]. In this respect, the beta values for H1 (0.44), H2 (0.38), and H4 (0.34) were positive and significant (*p* < 0.05 and all CIs had a nonzero interval). This statistical evidence was enough to statistically accept the theoretical statements of H1, H2, and H4.

Lastly, we evaluated the indirect effects to see the significance of mediating effects (H3: CSR→CE→ADOC = 0.29 and H5: CSR→CCI→ADOC = 0.22). To test the mediating effects, we enabled the bootstrapping option in AMOS software using a larger sample of 2000 bootstrap samples. The results of bootstrapping analysis were all significant, which confirmed that CE and CCI have significant mediating effects to explain ADOC. Hence, we confirmed that both CE and CCI are significant mediators between CSR and ADOC. Therefore, H3 and H5 were also accepted.

## 5. Discussion

The preliminary objective of this study was to investigate how CSR-related communication on social media can engage customers with a brand meaningfully by converting them into brand advocates. To this end, the statistical analysis revealed that a particular banking organization could effectively engage customers in CSR-related communication on social media by interactively communicating different CSR-related activities with them. Specifically, when customers see the CSR engagement of a banking organization on social media, they positively evaluate such noble intent of a certain bank, which not only provides them with a good reason to identify themselves with a bank strongly but also motivates customers to disseminate this socially responsible engagement of a bank to their social circles on social media including friends, peers, colleagues, and family members. Indeed, social media’s interactive, participative, and transparent atmosphere makes it an attractive communication forum for organizations to develop a meaningful customer–company relationship. Although the role of CSR from the standpoint of customers has been highlighted previously, this study meaningfully influences the debate on how engaging customers in CSR-related communication on social media can influence one important and less researched communicative behavior, ADOC.

Additionally, in line with the underpinning theory of this study (ST), the ethical commitment of a banking organization infuses positive emotions among customers. Hence, when customers observe CSR-related communication of a bank on social media, they not only positively evaluate such an ethical engagement of a bank, but they also pass on such information to others. As a corollary, customers are expected to strongly identify with a socially responsible bank and positively engage with such brands on social media. Earlier researchers have also argued in favor of this argument [48,49]. Hence, this study validates the theoretical statement of H1 by establishing that CSR-related communication of a brand on social media can determine ADOC. This is also in line with the studies of Castro-González et al. [47] and Kim et al. [112].

Similarly, to understand the underlying mechanism of how and why CSR influences different customers’ outcomes, including, but not limited to, ADOC, this study aimed to explore the role of customer emotions to explain the underlying logic of CSR–ADOC relationship. In this regard, this study highlights the mediating role of two important psychological factors CE and CCI. To this end, the statistically based evidence supported the theoretical statements of H3 and H5 by indicating a seminal role of customers’ emotions in influencing ADOC as an outcome of CSR. Specifically, this study is in line with the findings of Glavas [21] and Aguinis and Glavas [22] that the manifestation of different psychological factors (emotions, for example) as mediators provide justification for why CSR significantly drive certain behavior of individuals, including customers.

The available literature on CSR recognizes the potential benefits of CE to establish and sustain a meaningful competitive advantage. Moreover, an increasing body of research rates CE as an important predictor to influence an organization’s overall performance and business excellence. To this end, we align with past studies that contemporary customers expect businesses to be engaged in different CSR-related activities [69]. Especially with the rise of social media, contemporary customers tend to develop a social consciousness with businesses because they are interested to see how they are helping others by being the customers of a socially responsible brand. Additionally, customers are passionate about seeing how their relationship with a brand can help to create a better and sustainable future for this planet. To this end, when customers see that a certain banking institution shares different CSR-related communications on different social media forums, they are self-motivated to support such socially responsible brands by promoting it among their social circles. Hence, CSR influences the level of engagement on the part of customers, which then positively influences ADOC. This finding has also received support from previous researchers [24,57,72].

In the same vein, this study confirms the mediating role of CCI in explaining the CSR–ADOC relationship. The relationship marketing literature has recently given increasing traction to CCI for influencing different customer outcomes. Indeed, fostering CCI has become an important business imperative because research indicates that customers with an improved level of identification with a brand show a higher level of satisfaction and become loyal to a particular brand. At the same time, the existing body of research indicates that customers with a higher level of CCI are expected to promote a company toward excellence [28]. While the mediating role of CCI to influence customer behavior has been discussed previously, the mediating role of CCI to influence ADOC in a CSR framework on social media remained a less explored area. Therefore, this research influences this debate by highlighting the mediating role of CCI between CSR and ADOC on social media from the standpoint of the banking segment. More specifically, referring to ST, we argue that customers identify themselves with a socially responsible banking organization on the basis of its ethical engagement. The noble intent of an ethical bank as a part of CSR strategy motivates customers to develop strong CCI. In this vein, the increasing importance of social media has further facilitated the available literature on marketing communication to explain how customer engagement on social media with a brand influences their CCI [77,78]. Indeed, the emergence of social media enables customers to communicate effectively with a brand. The manifestation of social media has increased the effectiveness of CSR for a bank because by using social media as a communication forum, a specific bank can effectively engage customers in CSR-related communication, which improves their CCI and ultimately improves their ADOC. Thus, this study confirms the mediating role of CCI in influencing the CSR–ADOC relationship in a social media context. The early literature stream also confirms this finding [80,113,114].

### 5.1. Implications

#### 5.1.1. Implications for Theory

Our research significantly contributes to the available body of knowledge by offering important theoretical insights. First of all, our study is one of the sparse studies on the CSR–ADOC relationship. Although the existing body of knowledge highlights the important role of CSR in influencing customer behavior, e.g., loyalty intentions or purchase preferences, our research enriches this body of knowledge from an advocacy perspective, which previously remained an understudied area. Considering the seminal role of customers as brand advocates to the success of a brand, it was worthwhile to advance this debate in a CSR framework. Unlike some recent studies on CSR–advocacy behavior [47,115], our study advances the available literature on CSR in social media to meaningfully engage customers with a brand.

Second, this study is one of the pioneering studies in the domain of CSR, which attempts to highlight the seminal role of human emotions from a behavioral sciences perspective by proposing the dual mediating roles of CE and CCI to understand the underlying mechanism of why engaging customers in CSR related communication on social media can foster their advocacy behavior. This perspective, according to our best knowledge, was not previously highlighted. Considering the seminal role of human emotions to influence different behavioral intentions, especially advocacy, it was important to highlight the mediating mechanism of CCI and CE in a CSR–ADOC framework [116,117].

Third, most of the previous studies on CSR–customer behavior relationships were conducted in the context of developed countries [40,41]. This is important to enrich the existing literature from the perspective of developing countries because the cultural context nature of CSR requires separate investigations in the context of developing countries. This is important from this perspective. Furthermore, in developed countries, the notion of CSR relates well to the sustainability or environmental steps taken by an organization. However, in many developing countries, the concept of CSR relates to the philanthropic responsibility (charity and donations) of a business. Considering this context and the culture-specific nature of CSR, our study significantly advances the existing literature from a developing country’s perspective.

Lastly, this study advances the debate on the CSR–customer behavior relationship, especially from the banking segment of Pakistan. Theoretically, this investigation from the banking industry’s perspective is of critical importance because this segment operates under a tightly monitored standardized environment, making it difficult for a banking organization to differentiate itself meaningfully to survive in a stiff competitive environment. Considering the importance of customer referrals, recommendation, and endorsement for a brand on social media, it is really important to influence the debate of ADOC on social media in a CSR framework. However, previous investigations, especially from the standpoint of social media, were less focused on by other researchers.

#### 5.1.2. Implications for Practice

Our study makes some important policy implications. Firstly, in light of empirical evidence, our study highlights the seminal role of CSR-related communication in social media to positively influence ADOC. This finding is especially important for the banking segment, which faces the issue of competitive convergence due to a standardized service delivery pattern of this segment. In this respect, a particular bank can find a stable competitive base in the form of customers by converting them into advocates as an outcome of CSR. Customers feel elevated to be the buyers of a socially responsible organization because they believe that, by being the purchasers of an ethical brand, they are also contributing to the larger benefit of society and the biosphere.

Secondly, from an engagement perspective, enabling customers to participate in a bank’s CSR communication is something that engages customers on social media. Customers not only spread general awareness about an ethical bank, but they can drive participation and share public discourse in a more customer-driven manner which is considered more reliable than company-generated marketing communication. Customers’ engagement in CSR communication is also important for a bank to ensure the confidence of the public. For example, if a bank spends on community education as a part of its CSR strategy, it is not necessary that this would grab the attention of people significantly; however, when such actions of a bank are endorsed by the customers as social media users, it builds a good image of a particular bank in real sense.

Thirdly, another practical insight of our study is in highlighting the seminal role of customer emotions in influencing their behavior, especially ADOC. In this respect, the literature already acknowledges that emotionally charged customers bring various advantages to a certain brand. Customers with a higher level of emotional engagement with a certain brand are less likely to quit a brand; even in difficult times, they stand by their preferred brand due to their emotional relationship with such brands. To this end, our results indicate a role of customer emotions in the form of CE and CCI, which motivate them to strongly identify themselves with a brand due to its socially responsible behavior. Ultimately, the emotional pull generated by a brand due to its social responsibility engagement converts customers into brand advocates. Because social media provides an interactive communication forum, customers not only interact with a particular brand on social media but also tend to share their relationship experience with a brand in their social circles. Hence, by using social media as an effective CSR communication strategy, a certain banking organization can promote ADOC among customers.

### 5.2. Limitations and Future Research Suggestions

Considering the different significant theoretical and practical implications of this study, one should not mark this investigation error-free. Indeed, there are some potential limitations in this study which we want to highlight, along with possible suggestions for future researchers. First of all, this study collected data from two metropolitan cities in Pakistan. Due to different resources and time constraints, we did not include other cities in this survey; hence, the geographical concentration of this study is something that may serve as a potential limitation. For future studies, we recommend including more cities to deal with this limitation. Similarly, although the mentioned relations were all significant, we still feel that the theoretical framework of this study may be enriched by incorporating more variables as mediators and moderators. For example, in future investigations, it may be interesting to see how the role of admiration and gratitude as moderators or mediators may further enhance ADOC. Hence, it is suggested to include more variables in the theoretical framework of this study. Lastly, considering the context-specific and culture-specific nature of CSR, it is hard to generalize the results of this study to other segments of an economy and other cultures, although we believe that, in similar cultures such as Bangladesh or India, our study may reflect the same findings. However, we still recommend that future studies produce a comparison of different service segments and cultures. For example, comparing the hospitality sector with banking may be important in future studies because the hospitality sector also faces a similar challenge of competitive convergence to banking.

## 6. Conclusions

On the basis of the empirical results of this study, we suggest the banking segment of Pakistan to carefully plan and execute different CSR plans, especially from a marketing perspective. This is because well-planned and well-executed CSR strategies not only serve to highlight the philanthropic responsibility of a business, but such actions are also positively evaluated by the customers. Additionally, we suggest the banking institutions to align CSR plans with marketing communication by using different social media forums. When a particular bank announces its CSR-based initiatives on social media, customers respond positively and are expected to share the ethical perspective of a socially responsible bank within their social circles. Considering dynamic business environments, cutting-edge technologies, and many other factors, it is critical for a particular bank to develop a meaningful relationship with customers because, without the support of customers, it is almost impossible for a bank to be successful. At the same time, we recommend the banking organizations to incorporate the emotional perspective in their CSR communication strategies, because emotionally charged customers provide extra support to a particular brand they love. Moreover, emotional feelings with a brand create a strong social bonding between an organization and its customers. Similarly, as an outcome of CSR, positive emotions also significantly influence ADOC. To conclude, if rising rivalry and limited differentiation options are the biggest challenges in the banking segment, effective CSR communication on social media is a way forward for banking organizations.

## Figures and Tables

**Figure 1 behavsci-13-00032-f001:**
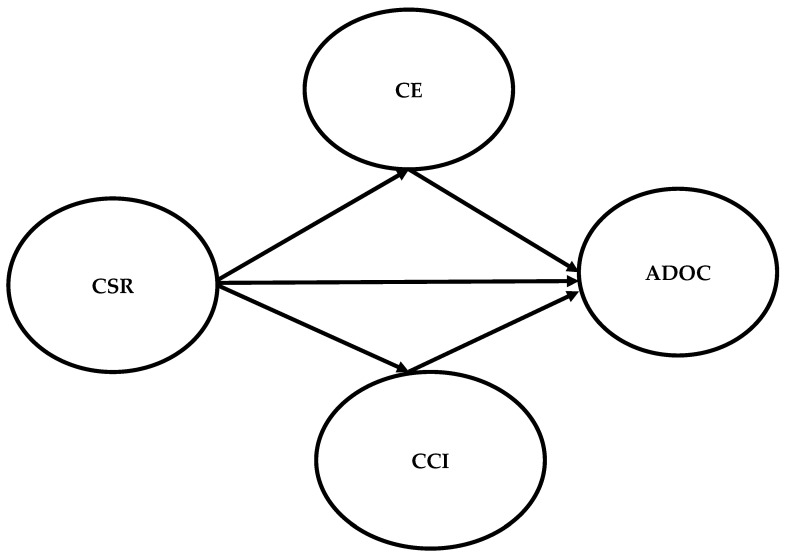
The conceptual model where CSR = corporate social responsibility, CE = customer engagement, CCI = customer company identification, and ADOC = advocacy behavior.

**Table 1 behavsci-13-00032-t001:** Data cleaning, outliers, and response rate.

Distributed	Returned	Unreturned	Removed	Outliers	Final
500 (HBL = 78) (ABL = 84) (MB = 67) (NBP = 95) (UBL = 64)	388	112	32	09	356
-	77.6%	22.4%	8.24%	28.12%	71.2%

**Table 2 behavsci-13-00032-t002:** Outliers.

Case No.	Mahalanobis d-Squared	p1	p2
243	12.733	0.002	0.000
9	8.450	0.015	0.006
64	8.450	0.015	0.002
119	8.450	0.015	0.001
175	8.450	0.015	0.000
229	8.450	0.015	0.000
20	7.679	0.022	0.001
75	7.679	0.022	0.000
130	7.679	0.022	0.000
243	12.733	0.002	0.000

**Table 3 behavsci-13-00032-t003:** Factor loadings of constructs.

	λ	λ^2^	E-Variance
CSR	0.79	0.62	0.38
AVE = 0.65	0.73	0.53	0.47
CR = 0.90	0.90	0.81	0.19
∑λ^2^ = 3.23	0.88	0.77	0.23
Total items = 5	0.70	0.49	0.51
CE	0.73	0.53	0.47
AVE = 0.61	0.77	0.59	0.41
CR = 0.86	0.89	0.79	0.21
∑λ^2^ = 2.42	0.71	0.50	0.50
Total items = 4	-	-	-
CCI	0.93	0.86	0.14
AVE = 0.63	0.69	0.48	0.52
CR = 0.90	0.74	0.55	0.45
∑λ^2^ = 3.17	0.76	0.58	0.42
Total items = 5	0.84	0.71	0.29
ADOC	0.78	0.61	0.39
AVE = 0.60	0.83	0.69	0.31
CR = 0.85	0.72	0.52	0.48
∑λ^2^ = 2.39	0.76	0.58	0.42
Total items = 4	-	-	-

Notes: λ = Item loadings, CR = composite reliability, ∑λ^2^ = sum of the square of item loadings, E-variance = error variance.

**Table 4 behavsci-13-00032-t004:** Model fitness.

Model	*χ*^2^/*df* (<3)	Δ*χ*^2^/*df* -	RMSEA (<0.08)	GFI (>0.9)	TLI (>0.9)	IFI (>0.9)	CFI (>0.9)
1	2.92	_	0.066	0.94	0.92	0.94	0.95
2	3.88	2.43	0.076	0.82	0.82	0.83	0.83
3	5.92	1.44	0.096	0.70	0.68	0.72	0.72
4	6.84	1.59	0.119	0.64	0.60	0.62	0.67

Note: Model 1 = four-factor model; model 2 = three-factor model combining CSR + CE into one factor; model 3 = two-factor model combining CSR + CE + CCI and ADOC; model 4 = one-factor model combining CSR + CE + CCI + ADOC.

**Table 5 behavsci-13-00032-t005:** Correlations and discriminant validity.

Variable	1	2	3	4
1	**0.80**	0.56	0.43	0.58
2	(3.29, 0.65)	**0.78**	0.39	0.52
3		(2.96, 0.53)	**0.80**	0.47
4			(3.49, 0.74)	**0.77**
				(2.99, 0.60)

Notes: Values in parenthesis = mean and standard deviation; bold values = discriminant validity; *p <* 0.001, 0.05. 1 = CSR, 2 = CE, 3 = CCI, 4 = ADOC.

**Table 6 behavsci-13-00032-t006:** Hypotheses results.

Hypotheses	Estimates (SE)	t and z Values	*p*-Value	CI	Decission
Direct effects					
H1: (CSR→ADOC)	0.44 (0.059)	7.46	****	0.22, 0.49	Supported
H2: (CSR→CE)	0.38 (0.052)	7.31	****	0.19, 0.42	Supported
H4: (CSR→CCI)	0.34 (0.043)	7.91	****	0.24, 0.65	Supported
Indirect effects					
H3: (CSR→CE→ADOC)	0.29 (0.036)	8.05	****	0.24, 0.82	Supported
H5: (CSR→CCI→ADOC)	0.22 (0.030)	7.33	****	0.36, 0.41	Supported
Total effect					
(CSR→ADOC)	0.61 (0.074)	8.24	****	0.39, 0.73	Supported

Notes: CI = 95% confidence interval with lower and upper limits. **** shows the level of significance.

## Data Availability

Data will be provided on demand.

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
