# Peer review of "The Relationship between Corporate Social Responsibility on Social Media and Brand Advocacy Behavior of Customers in the Banking Context"

_behavsci, 2022, doi:10.3390/bs13010032_

Round 1

Reviewer 1 Report

Manuscript Title: A robust model to develop meaningful customer brand relationship: a customer behavior perspective from banking services

Manuscript ID: behavsci-2065283

Journal: Behavioral Sciences

The paper is of greater importance pertaining to customer relationship management, brand advocacy and customer behavior as well as CSR.

Comments and Questions

Comment 1: The title could be modified as “Customer brand relationship: a customer behavior perspective in banking services” or “The relationship between corporate social responsibility and brand advocacy behavior of customers in the banking context”. Authors may also think about how to insert/ use the word “Social media” in the title as far as it is one of the main focuses of the study.

Comment 2: Abstract- Half of the abstract is dealing with the purpose or the objective of the study while giving little attention to the methods and materials as well as the major findings of the study. Hence, statements from lines 24- 29 may be removed and shifted to the introduction section. It is also strange to include several acronyms or abbreviations in the abstract. The authors may add the term “Brand advocacy” in the keywords.

Comment 3: The last paragraph of the introduction section (page 4, lines 168-177) shall be removed as long as it is a usual path for any research article. It has no relevance to spectators.

Comment 4: 2. Theoretical underpinning and literature review (line 178) - This can be framed as “Theoretical foundation and Literature review” or simply “Literature review” is sufficient.  

Comment 5: Figure 1 (Conceptual framework, page 7), the authors may draw an arrow that emanates from CSR and pass it to ADOC to examine the impact of corporate social responsibility on brand advocacy of customers’ behavior.

Comment 6: Please avoid redundancy of ideas or issues of CSR and Customer brand advocacy or engagement on social media (for instance look at pages 4 & 5, lines 199- 204), and page 13, lines 515-520.

Comment 7: The first two sentences of the abstract (see lines 22 & 23) is a direct copy-paste of the first two or three sentences of section 5 (conclusion, implications and limitations….see lines 575 & 576).

Comment 8. Implication for theory (section 5.1.1. page 15). I couldn't see new ideas other than the research gap already stated in the introduction and literature review in one or another way. It is a total reduplication with no relevance.

Comment 9. The authors may incorporate the list of acronyms and abbreviations in the last and attach the questionnaire in the appendix

Question 1: What do e1, e2, and e3 mean in the hypothesized framework?

Question 2: The authors selected the five sampled banks: HBL, MB, United bank limited (UBL), Allied bank limited (ABL) 338, and National bank of Pakistan (NBP) because of their CSR programs on social media. Do you mean that other banks are not being engaged in CRS? And, what are the CSR activities being implemented by the banks chosen as a sample of this research?

Question 3: How did the authors determine the sample size (i.e. 500)?

Editorial error- Below table 4, page 11, line 455, page 12 line 459, page 12 line 491, page 14, line 560 there is a word “betwixt”. Do you mean between?

Question 4. In your conclusion and implications, which CSR strategies and marketing communication approaches did you suggest to bank administrators? The conclusion is not clearly stated.  

Reviewer 2 Report

The paper deals with an interesting topic. However, the reviewer suggested a significant revision. Please see the followings. 

1. Title. the current title didn't reflect the key discussions in the paper. Authors are advised to revise the title to capture the essence of the paper, for example, CSR, brand advocacy. 

2. Abstract. "spark ADOC in CSR context" seems difficulty to comprehend. Please rephrase to make the meaning more accurate. What relationships are to be tested here? What research purpose is going to be achieved in the paper? Please revise. "difference banking customers" "different other implications" please specify which banking customers and implications you referred.

3. Introduction. A clear statement of research objective should be provided. The literature gap seems not well elaborated with strong arguments. Please add. Since social identity theory is the theory that supports the model, it is suggested to mention this theory in the introduction. 

4. Literature and hypotheses. "past research ...... pro-environmental behavior" sentences 206-209 should be revised. What individual behaviors should be changed because of CSR activities? Please specify. Additionally, H1, engaging customers in CSR-related communication on social media was not well discussed and defined in the hypotheses development statements. What engaging methods did you refer to? What specific CSR aspects did you refer to since CSR metrics are very broad? 

5. Figure. The constructs in the theoretical framework should be in circle shapes not in rectangle shapes since you define the constructs as latent constructs. Please revise. 

6. Method. "The major purpose of this study was to spark the debate on ADOC in the banking 320 context of a developing country, Pakistan. " sentences 320-321. This research objective statement is not consistent with the hypotheses and introduction. This makes readers very confused about the purpose of the research in the paper. Since the target banking services in the data collection were dominating banking systems in Pakistan, is it possible that customers are forced to choose the banking services because of limited choices they had? This might cause a biased response from the participants. Please justify. 

7. sentences 331, "investigate the relationships betwixt " typo, please correct. 

8. sentences 344-346, "To collect the data from banking customers, we devised a purpose-built strategy in 344 which the customers were randomly requested to fill out this survey questionnaire 345 (printed version following a paper-pencil method) when they were leaving a specific 346 branch of a bank or they were leaving an ATM facility. " The current data collection process makes the reviewer question about the random sampling approach claimed by the authors in the paper. Please provide more details about how randomization was conducted, how to define the population, sample, and scope. 

9. sentences 360-361, "We distributed 500 questionnaires among different banking custom-360 ers initially. Finally, " details are needed to explain what different banking customers were reached and collected in the final data. 

10. CSR-related communication on social media measures are missing, except a blur explanation. Please provide specific measurement items and explain the adaption process from the existing literature scale. 

11. factor loadings for each construct should be reported. 

12. Implications for practice should be extensively discussed with concrete practical suggestions and examples. 

Best of luck with research! 

Reviewer 3 Report

- what is qq.com? is it your lab's website? it seems not. please use the formal email of the authors' institutes.

- there are too many abbreviations in the abstract. try to use less.

- besides ADOC, how do you define and differentiate customers, consumers, operators, and users in the context of CSR? 

- besides social media, what are other factors relevant to this study? and why are they important? 

- line 274: H3. suggest discussing recent references on UX/UI of banking services/apps (from mdpi). such as: 

Zhu, D., Xu, Y., Ma, H., Liao, J., Sun, W., Chen, Y., & Liu, W. (2022). Building a three-level user experience (UX) measurement framework for mobile banking applications in a Chinese context: An analytic hierarchy process (AHP) analysis. Multimodal Technologies and Interaction6(9), 83. 

Cordeiro, T., & Weevers, I. (2016). Design is No Longer an Option–User Experience (UX) in FinTech. The FinTech Book: The Financial Technology Handbook for Investors, Entrepreneurs and Visionaries, 34-37.

- line 321: how can we borrow the experiences in Pakistan and spread them to other contexts and cultures? 

- make sure the numbers in tables 3 and 4 and 6 are correct. 

- can the authors provide specific examples of CSR in the research context? 

- line 496: it would be nice to discuss and summarize how the authors tested and proved the H1-5 (in a table).

Round 2

Reviewer 1 Report

You may carry out thorough proof reading to debug some mechanical errors. 

Reviewer 2 Report

The authors have successfully addressed the suggestions made by the reviewer. The reviewer doesn't have further suggestions. Congratulations on a great revision work!